



# Analysing flood fatalities in Vietnam using national disaster database and tree-based methods

Chinh Luu[1], Jason Von Meding[1], Sittimont Kanjanabootra[1]

[1]School of Architecture and Built Environment, University of Newcastle, Newcastle, NSW, 2308, Australia

*Correspondence to*: Chinh Luu (c3201999@uon.edu.au)

**Abstract.** Flood damage data recorded shows that Vietnam is vulnerable to flood hazards. This has severe consequences for the Vietnamese people, especially in terms of an unacceptably high death toll. To an extent, the high level of vulnerability is related to an insufficient capacity to cope with natural hazards, as is common in developing countries. On the other hand, social factors play their part and around the world, certain at-risk groups are systematically marginalised as a matter of

policy. The number of fatalities is the most important indicator in flood risk assessment. However, there is a significant lack of systematic research on flood fatalities in Vietnam. We respond to this gap and explore the national disaster database of Vietnam (DANA) with two objectives: (1) providing a comprehensive overview of flood fatalities in Vietnam, and (2) examining the damage-influencing variables (flood impacts) on flood fatalities. The tree-based methods were used for the exploration of influencing variables. Records stored in DANA show that over 14,927 persons were killed in floods in

Vietnam between 1989 and 2015 or at least 553 deaths and missing per year. The Mekong Delta region has the highest number of flood fatalities over the time period. However, South Central Coast and North Central Coast were the two most affected regions in flood fatalities historically when calculating an average per province per year in the regions. The analysis of tree-based methods shows that housing factor has the greatest influence on flood fatalities in Vietnam. The findings allow us to make recommendations for government policies on improving housing quality for the poor in flood-prone areas in

Vietnam.

## 1 Introduction

Asia is the most vulnerable continent to floods and storms according to the global disaster database as in Figure 1. There were about 3,620 floods and storms in Asia that resulted in over 8,085,516 fatalities on the continent between 1900 and 2016 (Figure 1). Vietnam located in South East Asia is one of the most vulnerable countries in the world to flood risk. Vietnam

recently ranks eighth in the most affected countries by extreme weather events between 1996 and 2015 (Kreft et al., 2016) and fourth among countries with the highest proportion of the population exposed to river flood risk worldwide (Luo et al., 2015).

Flood risk in Vietnam is as a result of monsoon tropical climatic characteristics, interlaced river systems, long coastline, and high population density in riverine and coastal areas (Razafindrabe et al., 2012; Chau et al., 2014b). The vast majority of





Vietnam's population lives in flood-prone areas, which exposes them to related hazards (Shaw, 2006). Previous research shows that developing countries suffer more catastrophic impacts than developed countries in disasters (Kahn, 2005; Toya and Skidmore, 2007; Hansson et al., 2008; Raschky, 2008; Peduzzi et al., 2009; Jongman et al., 2015).

Raschky (2008) explored global natural disaster damages between 1984 and 2004 and found that higher income countries predominantly record high economic loss but minimal loss of life due to disasters. Human loss of life is deemed the most important indicator in assessing flood risk (Maaskant et al., 2009). Several frameworks for estimating flood fatalities have been proposed based on flood characteristics. Penning-Rowsell et al. (2005) provided a framework to estimate injuries and deaths in floods using specific flood characteristics. Jonkman and Vrijling (2008) introduced a methodology to predict the flood mortalities including three steps of flood characteristics analysis, total exposed people estimation and mortality rate among exposed people assessment.

Studies on flood fatalities have been widely implemented in developed countries (Coates, 1999; Zhai et al., 2006; Ashley and Ashley, 2008; Jonkman and Vrijling, 2008; FitzGerald et al., 2010; Di Mauro and de Bruijn, 2012; Diakakis and Deligiannakis, 2013; Sharif et al., 2015), and to a lesser extent developing ones (Paul and Mahmood, 2016). The presence of research on flood fatalities in developing countries is minimal even though the rate of fatalities in these countries is most significant (Jonkman, 2005).

The most popular approaches in the research on flood fatalities are (1) the predictive models (Jonkman et al., 2002; Zhai et al., 2006; Jonkman and Vrijling, 2008; Di Mauro and de Bruijn, 2012; Di Mauro et al., 2012), and (2) the analysis of the causes of fatalities (Coates, 1999; Jonkman and Kelman, 2005; Ashley and Ashley, 2008; Jonkman et al., 2009; Sharif et al., 2015; Paul and Mahmood, 2016). These predictive models require a high level of detailed data (Jonkman and Vrijling, 2008). The analysis of the causes and circumstances are for specific regions or events.

Disaster damage data collection and analysis is gaining more and more attention (UNISDR, 2015). Storage and analysis of natural hazards impact data could provide basic information for decision-making and policy-setting in disaster risk reduction (Thieken et al., 2005; IRDR, 2014). Many sources of disaster loss and damage databases are available and summarised in documents of Simpson et al. (2014) and Grasso and Dilley (2013). For example, global disaster database launched by Centre for Research on the Epidemiology of Disasters (CRED) is available at Emergency Events Database (EM-DAT), http://www.em-dat.be; and 94 national disaster databases are stored and available at http://www.desinventar.net/DesInventar/index.jsp. Initiatives to compile the disaster databases at global and national scales are helpful to support disaster management information systems.

In Vietnam, the Damage Assessment and Needs Analysis (DANA) was developed to support the Vietnamese government to assess damage at the provincial and national levels with a focus on flood disaster (Bollin and Khanna, 2007; Below et al., 2010). Between 1999 and 2002, the Asian Disaster Preparedness Centre (ADPC) worked with USAID to review and strengthen a methodology for DANA (Bollin and Khanna, 2007). After that, CRED supported Vietnam to develop DANA. This organisation also help to develop the national disaster databases of other countries in Asia including Philippines, Bangladesh, Nepal, Indonesia and Sri Lanka at the same time (Below et al., 2010). Vietnamese government jointly





developed DANA through the Central Committee for Flood and Storm Control (CCFSC) (renamed Central Steering Committee for Natural Disaster Prevention and Control since 2017) (MARD, 2006; Hughey et al., 2011).

Several studies have investigated DANA including Nhu et al. (2011) and Hughey et al. (2011). Nhu et al. (2011) presented an overview of the impact of floods and storms Vietnam through a preliminary analysis of DANA database. Hughey et al.

(2011) examined the accuracy and completeness of the DANA procedures. To date, no research has explored DANA to analyse flood fatalities.

A systematic research on flood fatalities has so far not been undertaken for Vietnam. Therefore, we explored the national disaster database (DANA) with two objectives. The first objective is to comprehensively investigate flood fatalities in Vietnam. The second one is to examine the influencing variables related to flood fatalities using tree-based methods. We

applied tree-based methods including tree decision, bagging, random forests and boosting techniques to investigate the DANA data. The purpose is to explore the relative influence of independent predictors or variables related to flood fatalities based on the DANA dataset. The study contributes a method to analyse the national disaster database, provides a substantial insight in flood-related fatalities in Vietnam and offers a valuable application for other Asian countries.

Tree-based methods are supervised machine learning techniques. Tree-based methods are not only used for model prediction

but also for evaluating variable importance (Archer and Kimes, 2008; Strobl et al., 2008; Genuer et al., 2010). The methods have been used widely in many fields including genomics, proteomics, bioinformatics and other scientific fields (Bi, 2012). Recently they have been applied in hydrology studies (Ali et al., 2010; Carlisle et al., 2010; Loos and Elsenbeer, 2011) and flood risk studies (Merz et al., 2013; Spekkers et al., 2014; Chinh et al., 2015; Hasanzadeh Nafari et al., 2016; Wagenaar et al., 2017). A review of recent literature on the topic reveals that applications in flood risk studies are very recent. Four in five

articles used tree-based methods to select the substantial flood damage influencing parameters for different case studies using MATLAB software (Merz et al., 2013; Chinh et al., 2015; Hasanzadeh Nafari et al., 2016; Wagenaar et al., 2017). Spekkers et al. (2014) explored damage-influencing factors on insurance claims regarding water-related damage using decision-tree analysis and variable importance with statistical software R. There has not been any empirical study on the application of tree-based methods approach to analyse the damage-influencing parameters on flood fatalities.

The main advantages of tree-based methods are (1) useful in data exploration and interpretation (3) quantifying the importance of predictor variables, and (3) non-parametric method (Breiman, 2001; Strobl et al., 2008; Strobl et al., 2009). Tree-based methods were also shown to have more advantages than other machine learning algorithms (Breiman, 2001; Meyer et al., 2003; Auret and Aldrich, 2011) and traditional linear regression models (Peters et al., 2005). The main disadvantages of the methods are overfitting and little control over what the model does (Segal, 2004; James et al., 2013).

The overfitting problem can be avoided by implementing tree pruning technique (Merz et al., 2013). This technique involves cutting back a large tree to acquire a simpler tree. Tree-based approaches are potential to apply to the DANA data when the flood damage data includes many impact parameters or variables.



## 2 Disaster damage data

The flood damage data of Vietnam are collected in DANA database by the CCFSC (Hughey et al., 2011). The available period is from 1989 to 2015. The disaster loss data relates to the population, community infrastructure, public infrastructure, livelihood, and total economic loss. This substantial work has established a foundation for flood risk management in

Vietnam. Our study aims to explore DANA to generate a better understanding of flood risk related to human loss. The findings can provide a better rationale for decision-making related to disaster risk reduction.

The damage data in DANA is collected via a template with 12 main categories including impacts on humanitarian, housing, education, healthcare, agriculture, irrigation, transportation, fisheries, telecommunication, electricity, materials and economic loss. DANA only records the direct monetary damage for the recovery and reconstruction of damaged assets and

infrastructures, and it does not report the value of secondary losses such as business interruption (Wang et al., 2010).

Vietnam includes 58 provinces and five municipalities standing at the same level as provinces or 63 provinces. The government often gathers the various provinces into eight regions (in Table 1). The flood damage data was gathered from over 200 storm and flood data cards between 1989 and 2015. After that, the collected data was compiled to 63 provinces, and to 8 regions in Vietnam.

The damage data was collected from 1989 to 2015. Each year was considered an observation. The data included 27 samples (27 years between 1989 and 2015) for 63 provinces, so there were 1701 observations. The flood fatalities or humanitarian impact ($X1a$ variable) was set as an outcome or a dependent variable (as in Table 2). The impacts from $X2$ to $X11$ were set as independent variables (Table 2). Ten predictors from $X2$ to $X11$ and one dependent variable $X1a$ were used as input for the tree-based methods analysis.

The observed data is flood damage data, so it is random and contains many zero values. Some observations contain all zero values if no storms or floods occurred. The histograms of variables are not a normal distribution. Therefore, data transformation techniques need to be conducted. Logarithm function method for both dependent and independent variables was applied to data transformation. The logarithm function was selected after a series of testing with all types of data transformation techniques. The histogram and linearity checking was used to choose the most suitable transformation

function.

The correlations between variables were investigated using the statistical software R (R Core Team, 2016) with '*corrplot*' package (Wei and Simko, 2016) and '*car*' package (Fox et al., 2016). Figure 2 showed the correlation coefficients of 10 predictors ($lgX2$ to $lgX11$) and a dependent outcome (flood fatalities, $lgX1a$) after data transformation. The result revealed that the dataset has highly correlated variables, such as 0.72 between housing impacts ($lgX2$) and fatalities ($lgX1a$). Random

forests are a suitable tool for this type of data (Strobl et al., 2008). The scatter plot between housing impacts ($lgX2$) and fatalities ($lgX1a$) was illustrated in Figure 3. It was quickly evident in the data that housing impacts are closely related to flood fatalities.





## 3 An overview of flood fatalities in Vietnam

Vietnam is extremely vulnerable to floods. It is located in the monsoon tropics, with over 3,450 rivers and streams, and a 3,260 km long coastline. Flooding is caused by rising water level in rivers during storms or heavy rains or by typhoons in coastal areas. The rainy season or storm season lasts from May to October in the Northern part, from June to December in

the Central, and from July to December in the South (Vu et al., 2015). Many inhabitants live along rivers and coasts, and their livelihoods depend heavily on the natural environment. Flood events are more likely to affect low-income communities. There were more than 295 major storms (Level 6 to 12) to impact Vietnam the period 1961 to 2014  (NHMS, 2015) as shown in Figure 4. The number of annual occurring storms is increasing. About five main storms occurred per year during the 1960s, 1970s, 1980s and 1990s. However, it was up to over seven main storms each year during the 2000s (Figure 4).

EM-DAT defined deaths or fatalities to include persons confirmed as dead and missing persons presumed dead (Below et al., 2010). The death toll is the most important indicator in flood risk assessment (Maaskant et al., 2009). The statistical software R (R Core Team, 2016) was used to examine the national disaster database (DANA) of Vietnam. Between 1989 and 2015, floods killed over 14,927 people and injured 16,829 people (Figure 5). Alternatively, floods resulted in at least 553 deaths and 623 injured people per year in Vietnam. Typhoon Linda in 1997 alone killed more than 3,000 people (Figure 5).

Vietnam comprises three main regions, the North, the Central and the South (Table 1). Each region in Vietnam has its own flood risk due to topography conditions. In the North, floods are effectively controlled by five large hydropower plants on Da river basin. The main causes of flooding in the South are heavy rains, storms, tides and sea level rise (Wassmann et al., 2004). Also, community livelihoods in the South are affected by climate change and hydropower development in the Upper Mekong in China, Laos, Cambodia and Thailand regarding fisheries, sediment decline and drought (Kummu and Varis,

2007; Barlow et al., 2008). The Central region is characterised by a steep and narrow terrain, and fragmented by rivers deriving from westward mountain ranges to the East Sea of Vietnam. Due to such conditions, this region is often exposed to floods and storms causing significant losses in human lives and properties (Shaw, 2006; Tran et al., 2008; Chau et al., 2014a).

Vietnam government divided the three main regions into eight regions as in Table 1. It includes Northwest (NW), Northeast

(NE), Red River Delta (RRD), North Central Coast (NCC), South Central Coast (SCC), Central Highlands (CH), Southeast (SE) and Mekong Delta (MD). The greatest numbers of fatalities occurred in MD and SCC regions respectively with over 4,000 people killed over the 27-year period (Figure 6). If calculating an average per province per year in the regions, SCC and NCC were the two most vulnerable regions historically with 19 and 16 mortalities respectively (Figure 7). The SE and RRD region had the lowest mortality with 362 and 409 respectively in the last 27 years (Figure 6) or an average per province

30   per year of 2 fatalities (Figure 7).

Spatial patterns of flood fatalities, injured and casualties (fatalities and injured) for 63 provinces in Vietnam was created using the compiled DANA data and ArcGIS 10.1 software (Environmental Systems Research Institute Inc., USA) in Figure 8. The three provinces, Ca Mau, Quang Nam and Quang Ngai, had over 800 fatalities during the observed period. The nine



provinces had from 401 to 800 deaths during the recorded period, including Thanh Hoa, Nghe An, Thua Thien Hue (in NCC), Da Nang, Binh Dinh, Khanh Hoa (in SCC), and Kien Giang, An Giang, Dong Thap (in MD). Seven provinces recorded over 800 injured people including Thanh Hoa, Nghe An, Quang Binh, Quang Tri, Quang Nam, Quang Ngai and Ba Ria-Vung Tau. Cau Mau, Quang Nam and Quang Ngai provinces had the highest numbers of both flood casualties and fatalities.

## 4 Tree-based methods

Tree-based methods are supervised learning algorithms. The methodology of these methods is based on classification and regression tree (CART) of Breiman et al. (1984). The methods construct a multitude of decision trees and select the best of the group to be used for constructing predictive models (Liaw and Wiener, 2002). Tree-based methods include decision trees (regression and classification trees), bagging, random forests and boosting techniques. Decision trees are useful for visualisation and interpretation. Bagging, random forests and boosting techniques are used to improve the prediction accuracy of tree-based methods (James et al., 2013).

Tree-based methods are not only used for model prediction but also to measure the importance of variables with random forests and boosted trees (Auret and Aldrich, 2011). The variable importance measurement was applied to the DANA database to analyse the influencing variables (flood impacts) relating to flood fatalities. Flood humanitarian loss was set as a dependent outcome. Other flood impacts including housing, education, healthcare, agriculture, irrigation, transportation, fisheries, telecommunication, electricity and materials, were set as independent predictors. The relative importance of predictors was used to investigate the influencing determinants or factors on flood fatalities in Vietnam.

Cross-validation was performed to valid the data in this study. First, the original samples were randomly divided into two groups, training and testing data sets with equal size. Second, a model was developed in the training dataset. Third, the model was validated using the testing dataset. Finally, an index, Mean Squared Error (MSE), was used to evaluate the performance of models. The cross-validation procedure was undertaken to ensure that the parameter estimation and model generation of regression trees, bagging, random forests and boosting are entirely independent of the test data. Each of the groups was chosen one by one for assessment, and the cross-validation checked the MSE index.

In the next paragraphs, we present some detail underpinning these techniques.

### 4.1 Regression trees

Regression trees are possible for continuous variables. This method aims to develop an analysis that can be used to predict the outcome. The regression trees divide the predictor variables into distinct and non-overlapping regions. The splitting process continues until a stopping criterion is met. The tree is split to attain lower variance and better interpretation (James et al., 2013). Therefore, Tree Pruning technique is used to fit the regression trees.



## 4.2 Bagging and Random Forests

The regression trees (decision trees) suffer from high variance (James et al., 2013). To construct a more powerful prediction model or to obtain a low-variance model, the bagging or bootstrap aggregating technique is applied. Bagging is used to create bootstrap samples from the training data. Bagging can be considered the special case of random forests, which occurs when the random samples are equal the number of predictors (Breiman, 1996). The construction of a random forest algorithm is as follows (Liaw and Wiener, 2002; Archer and Kimes, 2008):

1. The number of observation in the data set is $n$
2. A sample of these $n$ is taken randomly but with replacement
3. If there are $m_{try}$ predictor variables, $m$ variables (m < $m_{try}$) are randomly chosen from $m_{try}$ at each node. The best split on these $m$ variables is used to divide the node
4. Each tree is developed to the largest extent possible, and no pruning
5. New data is predicted by combining the predictions of the $n_{tree}$ trees

A performance evaluation of a prediction algorithm should be made using independent test data sets that were not employed in a training data set. Some types of cross-validation methods are usually used to evaluate the performance of models including Out of bag error (OOB) estimate and Leave-one-out cross-validation (LOOCV). The OOB error estimate is accurate provided that enough trees have been grown (Liaw and Wiener, 2002), and is used for performance evaluation in this study.

One of the most powerful tools of random forests is to measure variable importance, which is of interest in various applications. The variable importance can be measured in the random forests by Gini importance, permutation importance or raw importance over all trees.

## 4.3 Boosting

Boosting technique is another approach for improving the accuracy of a decision tree result. Boosting works in the same way of bagging in creating bootstrap samples from the training data, however, the trees are grown sequentially (James et al., 2013). Boosting constructs many smaller trees. Each new tree in this technique works to adjust the defects of the current ensemble.

## 5 Application of tree-based methods in the study

The aim of tree-based methods is to find the relative influence of independent variables (as in Table 2) on the humanitarian impacts (flood fatalities).

## 5.1 Regression trees

The DANA database was explored to examine the influence of the ten main direct flood impacts (variables X2 to X11 in



Table 2) on flood fatalities (variable X1a). The regression trees were drawn and analysed using the statistical software R (R Core Team 2016) with three packages: '*rpart*' (Therneau et al., 2015), '*rattle*' (Williams, 2011) and '*rpart.plot*' (Milborrow, 2016). Figure 9 showed a fitting regression tree to the data. The root node is housing impacts (*lgX2*), and the second critical node is transportation impacts (*lgX7*). The tree indicated that the housing impacts (*lgX2*) is the most significant predictor.

Cross-validation method was operated to select the most accurate tree-based technique and to check the data validation. The data was split into equal-sized training and testing datasets from 1701 observations. The Mean Square Error (MSE) index was used to assess the performance of the regression model. The MSE of the test set of the regression trees is 0.81. The square root of the MSE is 0.9. The MSE was compared among techniques in next sub-sections.

## 5.2 Bagging and Random Forests

Bagging algorithm was implemented by the '*randomForest*' package (Liaw and Wiener, 2002) in statistical software R with the steps in Section 4.2. The bagging technique was applied for ten predictors and an outcome. All predictors were considered at each split of the tree with 500 trees. The result shows that mean of squared residuals is 0.62; the percentage of variable explained is 62.8%. The cross-validation was also executed to validate the data. The data was divided into two folds, train and test datasets with approximately equal sizes (850, 851) of 1701 observations. The mean square error (MSE)

of the test set is calculated as 0.73. The square root of the MSE is 0.85. This result is a bit lower than that obtained using the regression tree (0.9) in section 5.1.

The mean of squared residuals and percentage of variance explained are based on OOB error estimate. The results of OOB and test error in Figure 10 were estimated with a function of $m_{try}$. The $m_{try}$ is the number of selected variables at each split of each tree in the tree regression model, varies from 1 to 10. Figure 10 showed that the MSE indexes for $m_{try}$ between 1 and 10

of both OOB error and test error are low and around 0.6 to 0.7. The blue curve of test error is a little bit of lower than the red curve of OOB. The OOB is performed on the train dataset and different with the test error. The error estimate is much correlated when the random forests with $m_{try}$ from 1 to 10 are very similar each other. The red curve is quite smoothly above the blue curve. The $m_{try} = 3$ seems to be about the best for both OOB and test errors. However, $m_{try}$ can be selected from 1 to 10 since the difference of MSE is quite small.

The random forests process is grown as the same way of the bagging with using the lower random samples $m_{try}$. The $m_{try} = $ p/3 is by default of a random forest of regression trees. Here we used $m_{try} = 3$. The result specified that mean of squared residuals is 0.59, and percentage of variable explained is 64.5%. The test set MSE is 0.70. The result indicated that random forests offered an enhancement over bagging in this case.

The raw importance over all trees of the random forests is shown in Figure 11. Housing impacts (*logX2*) is the highest

importance variable to the fatalities and accounts for the major component in the model.

## 5.3 Boosting

The '*gbm*' package (Ridgeway, 2015) in statistical software R (R Core Team, 2016) was used to fit boosted regression trees



for the DANA dataset. We also used cross-validation method to check the performance of boosting technique. The test MSE is 0.70, which is similar to MSE of random forests and better to those of bagging and regression trees.

The result in Figure 12 suggested that the housing impacts variable has the most influence on the regression tree model of fatalities, followed by transportation impacts. It is consistent with the results of random forests and regression trees techniques. Figure 13 displayed the plots of partial dependence for the two most importance variables. The plots illustrated the marginal influence of the selected variables on the outcome variable (flood fatalities) after integrating other variables. In this case, the number of death toll increases with both housing impacts and transportation impacts.

## 6 Discussion

It is essential to record and analyse natural hazards impacts to provide the basic information for decision-making and policy-setting in disaster risk reduction (IRDR, 2014). These analyses can consolidate the first priority of Sendai Framework for Disaster Risk Reduction, understanding disaster risk which is the first step towards disaster risk reduction (UNISDR, 2015). The CCFSC of Vietnam has documented the extensive range of flood impacts from humanitarian loss (fatalities and missing people) to agriculture, housing, infrastructure damages and economic losses from 1989 to 2015 through DANA database. This database can provide necessary information for flood risk management activities in Vietnam when the DANA data are analysed and explored in details. This study aims to investigate DANA for better understanding the humanitarian impacts of flood risk in Vietnam.

### 6.1 Flood fatalities in Vietnam

Many structural and non-structural measures for preventing or mitigating flood risk have been implemented in Vietnam such as flood control dyke systems, reservoir dams, construction of flood resistant housing, upstream forest protection, and enhanced early warning systems. However, the existing disaster data shows that floods continue to severely affect the Vietnamese people, particularly if we consider the unacceptably high death toll of over 14,927 fatalities between 1989 and 2015.

Flood fatalities have been rare in developed countries (Crichton, 2004; Mojtahedi and Oo, 2016). Dittmann (1994) found that there were 119 flood deaths per year in 1959-1991 for the United States. Meanwhile, the flood damage records of Vietnam from 1989 to 2015 indicated that at least 14,927 deaths and 16,829 injured people in floods. This loss of life remains very high in Vietnam. The vulnerability is continuing in 2016 with 111 deaths and missing, and 121 injured in flood events in November and December in Central Vietnam as a report from UN Country Team in Vietnam (2016).

Figure 5, Figure 6, Figure 7 and Figure 8 presented an overview of flood risk in regions and provinces of Vietnam. Our results show that the South Central Coast, North Central Coast and Mekong Delta are the most affected regions by flood risk on humanitarian, followed by North West and North East. The South East, Red River Delta and Central Highlands have the lowest number of flood fatalities in Vietnam. The most affected regions are coastal areas and lowland areas.





Figure 8 showed spatial patterns of the flood casualties for 63 provinces in Vietnam. The provinces along the Central coastal regions are most vulnerable to flood casualties. In fact, high annual deforestation rates occurring at upstream areas, especially in the Central Highlands (McElwee, 2004) invariably lead to flooding vulnerability increase at the downstream or low-land areas. The map can provide information for flood risk managers to identify the high-risk areas and to take

appropriate flood risk mitigation measures.

It still lacks proactive approaches to flood risk management in Vietnam (Chau et al., 2014b). Deaths occurring during floods are almost due to the passive response of flood risk managers and populations to flood events. Flood risk management activities at local levels must emphasise on mitigation and preparedness activities before flood events happen.

## 6.2 Tree-based methods in application

Tree-based algorithms are one of the best supervised machine learning methods, providing powerful predictive models with high accuracy, stability and ease of interpretation (Breiman, 2001). The objective of tree-based methods in this study was to analyse flood fatalities in Vietnam through variable importance assessment using the national disaster database, DANA. The recorded DANA data includes a broad range of flood impacts from human loss (death and missing people) to agriculture, housing, infrastructure damages and economic losses between 1989 and 2015.

The tree-based techniques including decision trees, bagging, random forests and boosting were implemented to explore the DANA data for a better understanding of fatalities related flood hazards. The fatalities or humanitarian impact was set as a dependent variable, and the other impacts were set as independent variables. The cross-validation index (MSE) and results among the three techniques were compared and assessed to have the most accurate model and result.

The regression tree in Figure 9 provided an overview of influencing variables on flood fatalities. The most influencing

variable is housing impacts ($lgX2$), followed by transportation impacts ($lgX7$). This tree displayed a useful interpretation of variables used in the analysis. After that, bagging, random forests and boosting techniques were applied respectively to improve the prediction accuracy of the model.

The tree-based models were validated by cross-validation method with MSE checking. The MSEs are 0.81, 0.73, 0.70 and 0.70 for regression trees, bagging, random forests and boosting techniques respectively. It indicated that boosting and

random forests yielded the improvement over the regression trees, bagging techniques. Also, an out of bag error estimate, OOB, was used to evaluate the performance of the regression tree model in Figure 10. The results showed that the tree-based models were validated and applicable.

The analysis of tree-based methods specified that housing impacts had the greatest influence on flood fatalities in Vietnam (Figure 9, Figure 11 and Figure 12). To summarise, the importance of damage-influencing variables to flood fatalities

ranging from high to low: housing, transportation, agriculture, fisheries, irrigation, healthcare, education, electricity, telecommunication, and materials impacts respectively (Figure 12). However, housing impacts attribute accounts for the majority of influence on flood fatalities at 59.93% (Figure 12). The higher the housing impacts are in floods, the higher the



fatalities are (Figure 13). The reality of Vietnam, with over 70% of the population living in rural areas with limited resources help to explain this result.

The houses of farmers are mostly single-storey and in a poor condition. The structure of these houses is not strong enough to withstand the forces of storms and floods as examples in Figure 14 and Figure 15. The people living in such houses that are most likely to be killed by drowning. De Bruijn and Klijn (2009) asserted that people might also die after becoming trapped in a house collapsing during floods. In addition, high flood depth levels threat the lives of inhabitants in single-storey houses when they have no room to evacuate as in Figure 15.

Almost all flood affected people in Vietnam are part of poor or marginalised groups. They often face grave difficulties in disaster recovery. The affected households only receive a minuscule amount of recovery support. For example, they can receive the support of three million Vietnam Dong (equivalent to 150 USD) for a death or missing person, and one million Vietnam Dong (equivalent to 50 USD) for an injured person stipulated in the Decree 67/2007/ND-CP of government. There is not insurance scheme for the human impact of natural hazards in Vietnam.

The analysis in this study was based on the available data in the national disaster database, DANA, so it was limited to the analysis of influencing damage-factors on fatalities. We call for more detailed research into the cause of flood fatalities in Vietnam. Further research would benefit if the humanitarian disaster impacts in DANA were documented with more details such as reason of mortalities, genders and ages.

**7 Conclusion**

This study presents a systematic research on flood fatalities in Vietnam through exploring the DANA national disaster database. It comprises an overview of flood fatalities in the whole country and an analysis of the influencing variables on flood fatalities. Tree-based approaches are used to identify the most significant damage-influencing variables or attributes to flood fatalities.

To our best knowledge, there have been no long-term empirical studies on the application of damage-influencing factor analysis using machine learning algorithms in investigating flood fatalities. This paper proposed an approach to analyse the national disaster database of Vietnam and explore the damage-influencing factors relating to flood fatalities. The analysis can add some information for policy makers and decision makers in flood risk management to take appropriate measures and government interventions to reduce the humanitarian impacts of flood hazards.

There has not been any previous systematic study on flood fatalities in Vietnam. This study can contribute to the body of flood hazard knowledge by analysing and reporting on flood fatalities in Vietnam. The analysis of indicators related to flood fatalities can provide additional information for public awareness programs and public safety enforcement activities. This approach can be used to analyse the damage-influencing factors to flood fatalities using the flood damage data collections which are observed over an extended period and categorised by attribute in other countries or regions.



Finally, we would like to make three main recommendations for flood risk management activities in Vietnam. First, the disaster database documenting should include more details on the cause of deaths, gender and ages. Second, government policies should draw more attention to the improvement of housing quality for the poor in flood-prone areas. Lastly, flood risk management activities should shift the focus to a proactive approach, including mitigation and preparedness.

**Acknowledgments**

We wish to thank the Central Steering Committee for Natural Disaster Prevention and Control in Vietnam for providing us with the invaluable dataset. Also, Chinh Luu acknowledges the University of Newcastle International Postgraduate Research Scholarship for her research.

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



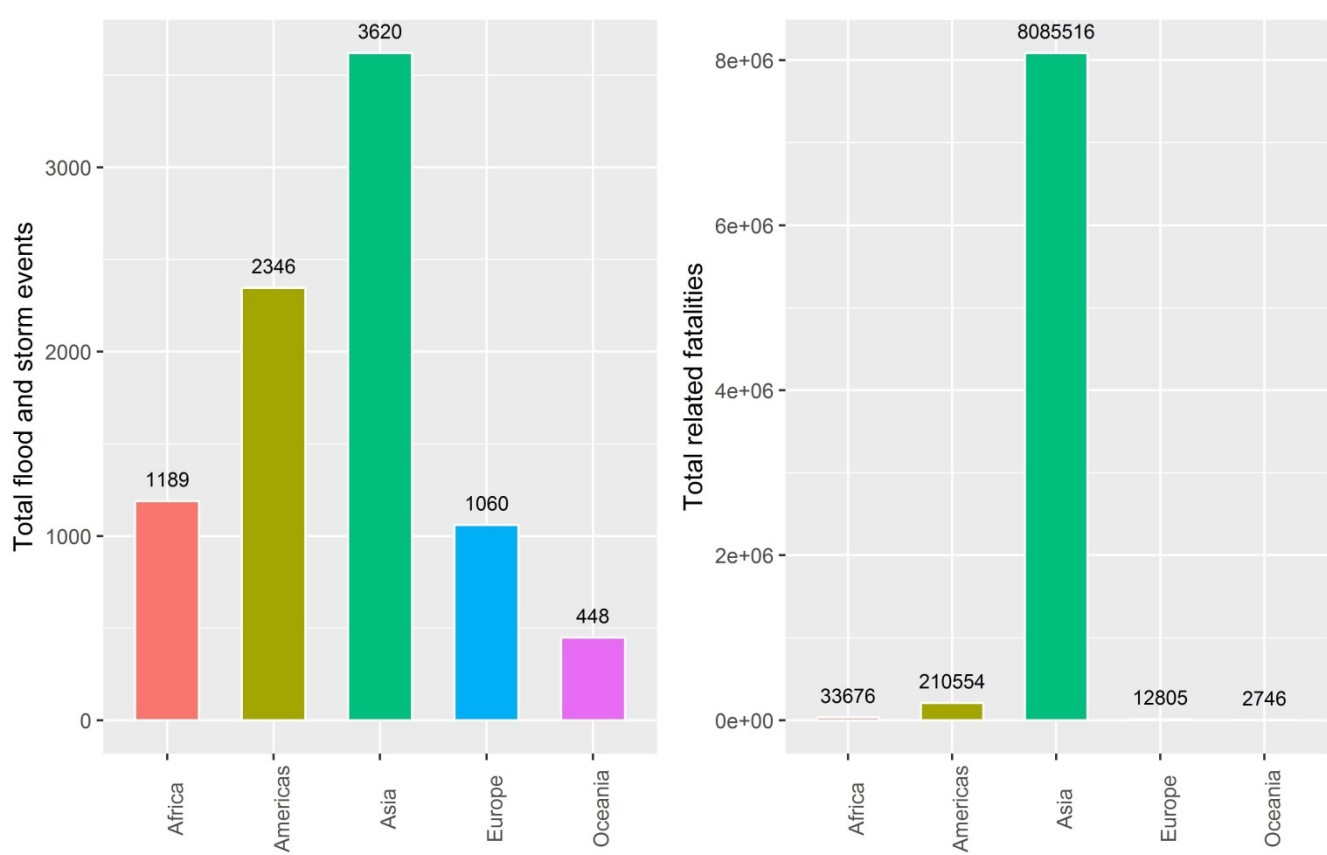

**Figure 1: Global flood and storm events and related fatalities between 1990 and 2016 (data source: http://emdat.be).**





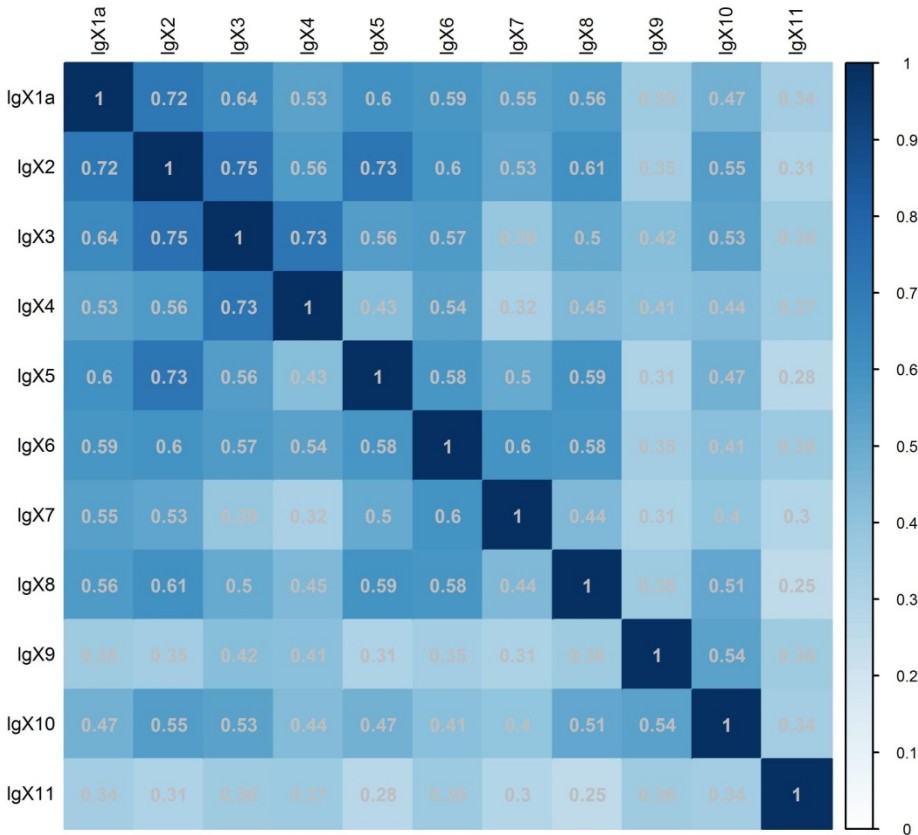

**Figure 2: Correlation coefficients of 11 variables (10 predictors and flood fatalities) on transformed data.**





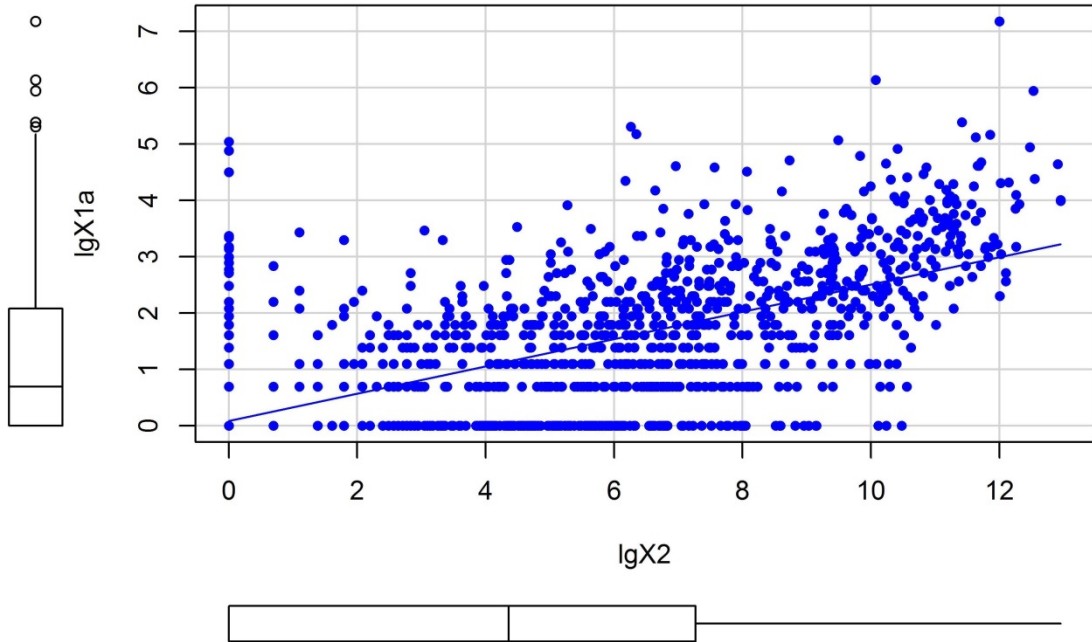

**Figure 3: Scatter plot showing the relationship between flood fatalities (lgX1a) and flood housing impacts (lgX2) (transformed data).**




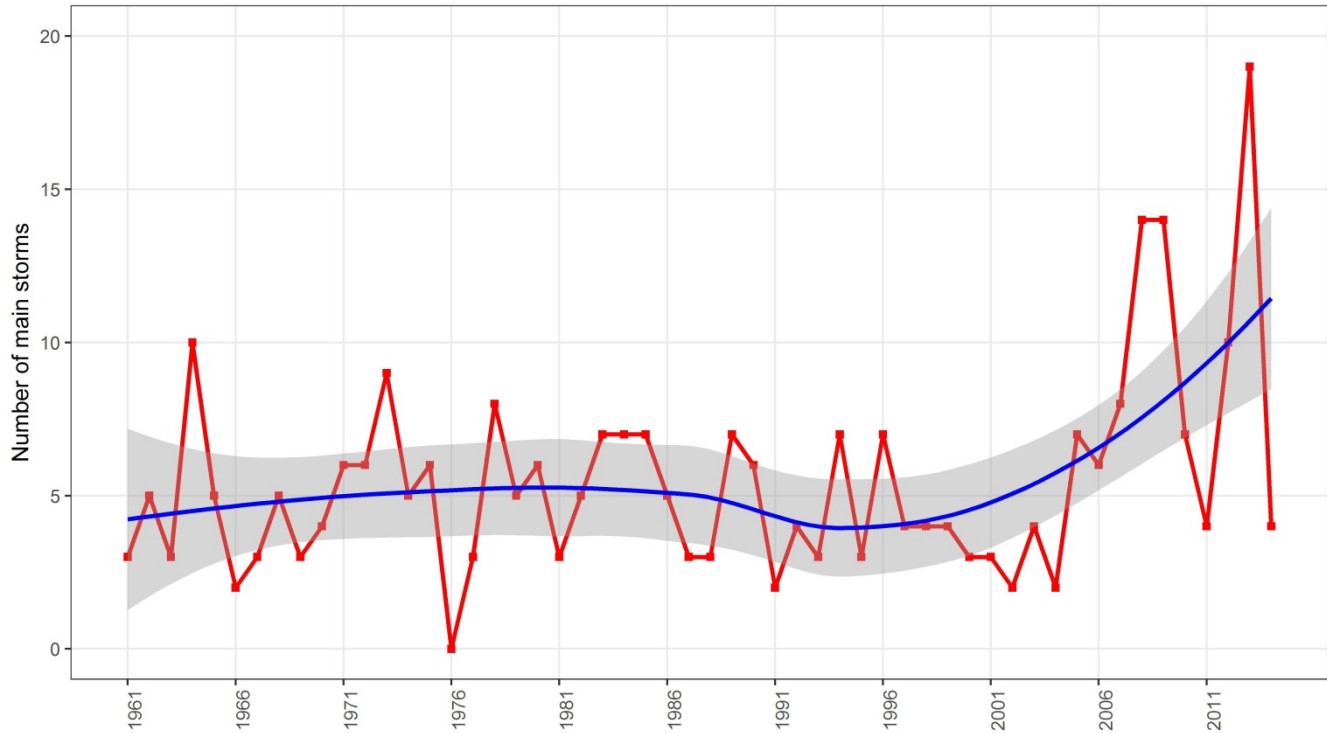

**Figure 4: Number of major storms (Levels from 6 to 12) landed in Vietnam between 1961 and 2014 (Source: compiled from NHMS (2015)). (Note: the storm level is determined by an average wind speed. Levels 6, 7, 8, 9, 10, 11, and 12 have average wind speed of 39 to 49 km/h, 50 to 61 km/h, 62 to 74 km/h, 75 to 88 km/h, 89 to 102 km/h, 103 to 107 km/h, and 118 to 133 km/h respectively).**





**Figure 5: Total fatalities and injured people caused by floods between 1989 and 2015.**





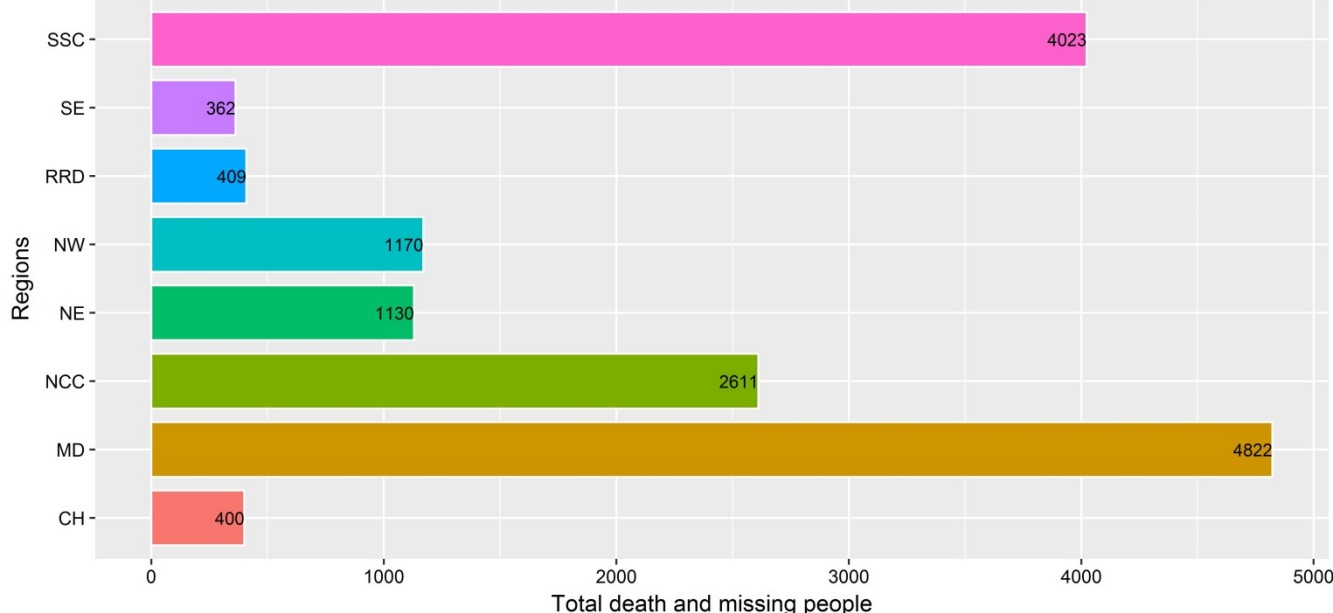

**Figure 6: Total death and missing people in regions in Vietnam between 1989 and 2015. (Note: Northwest (NE), Northeast (NE), Red River Delta (RRD), North Central Coast (NCC), South Central Coast (SCC), Central Highlands (CH), Southeast (SE), Mekong Delta (MD)).**





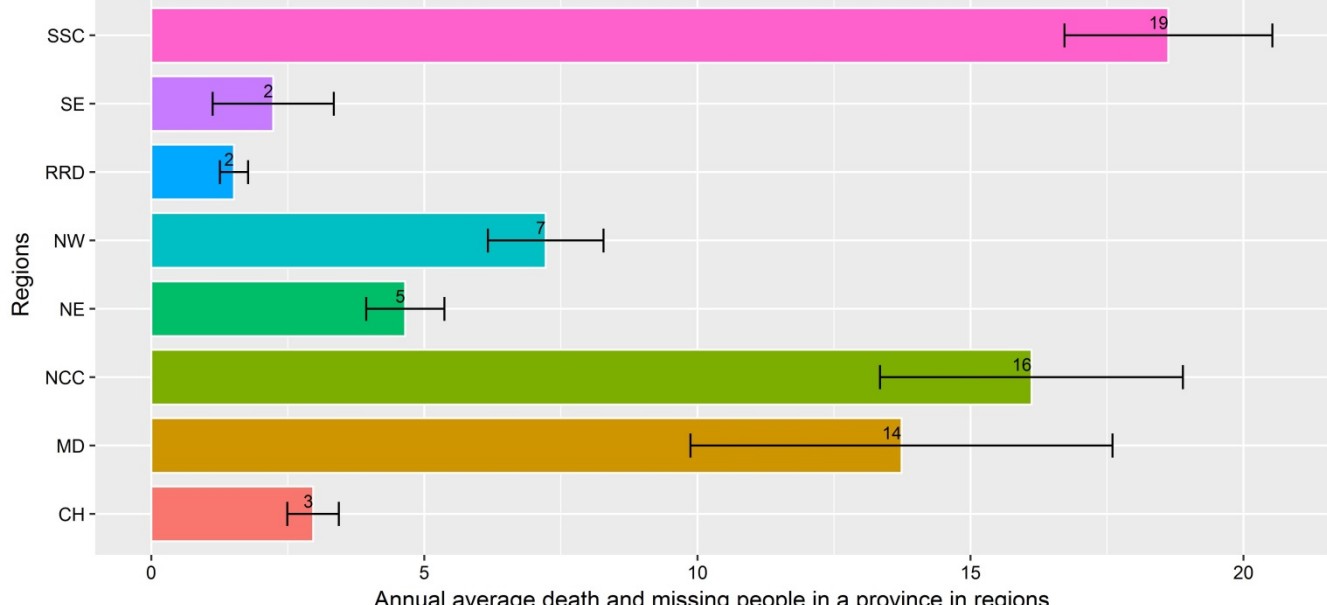

**Figure 7: Annual average and standard errors of flood fatalities in a province in regions in Vietnam between 1989 and 2015.**



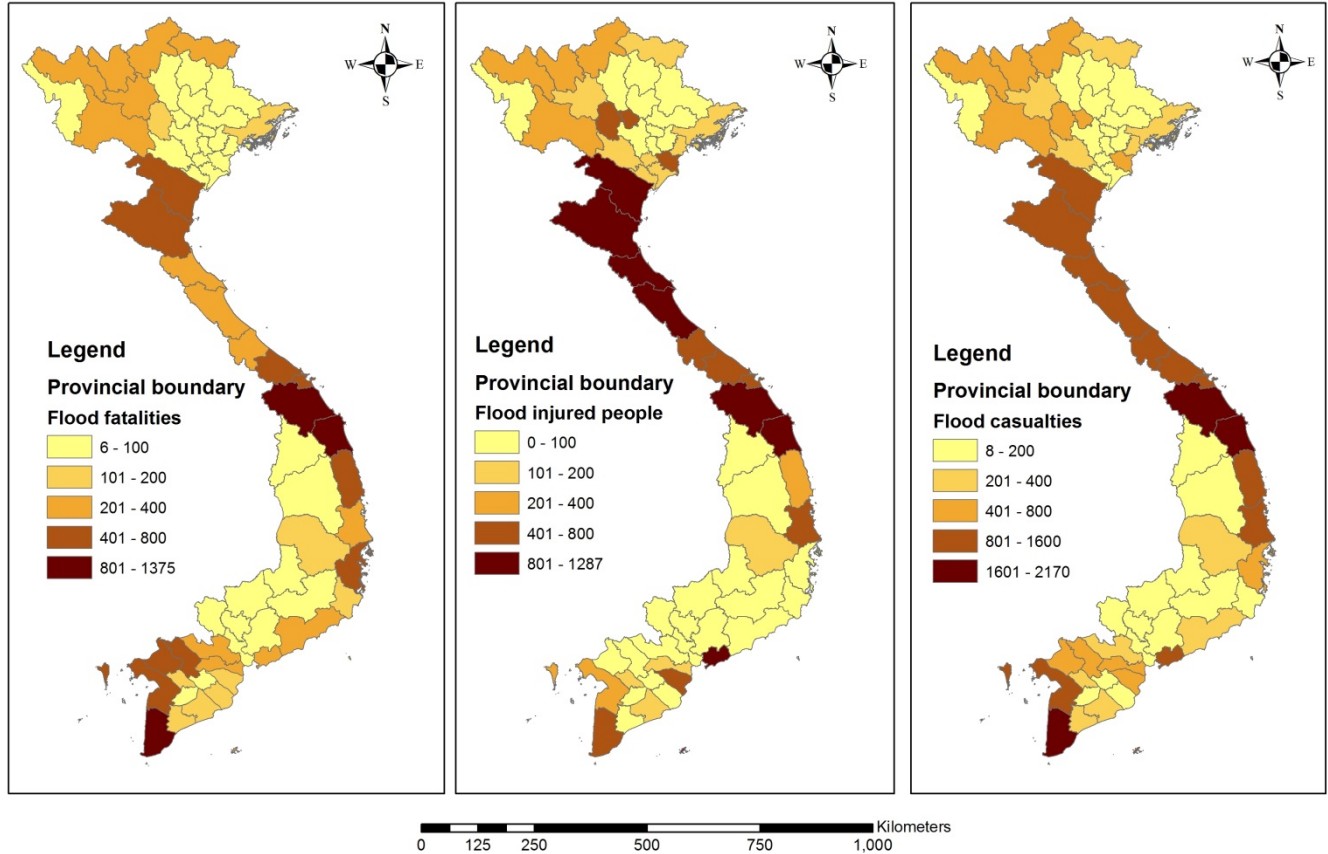

**Figure 8: Spatial patterns of flood-related fatalities, injured and casualties (fatalities and injured) by provinces in Vietnam in the period between 1989 and 2015.**


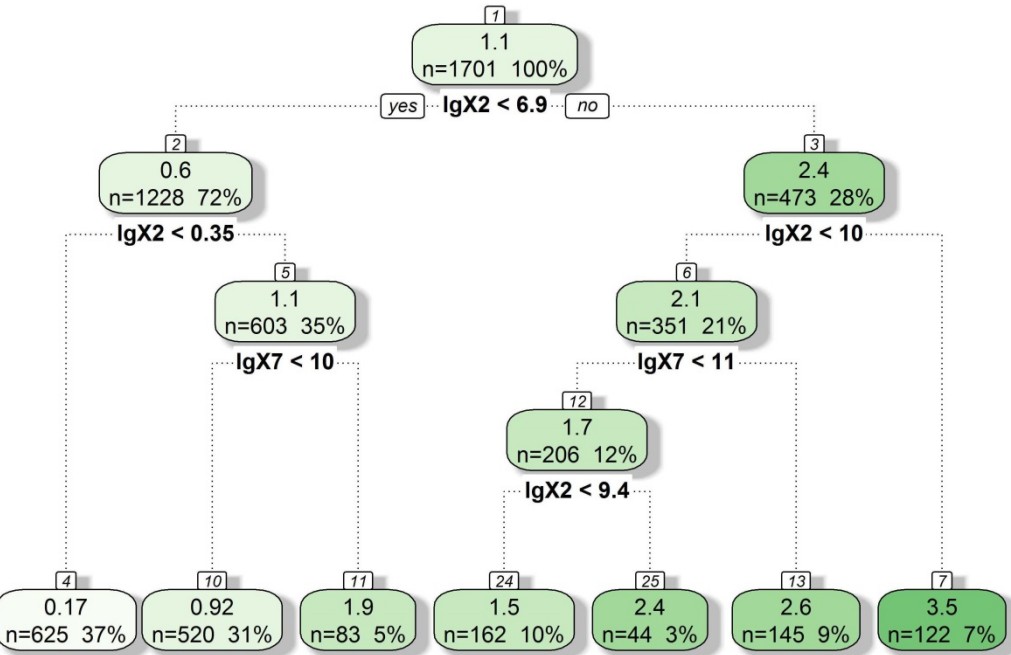

**Figure 9: Regression tree for estimating flood fatalities, lgX2: housing impacts, lgX7: transportation impacts (transformed data).**




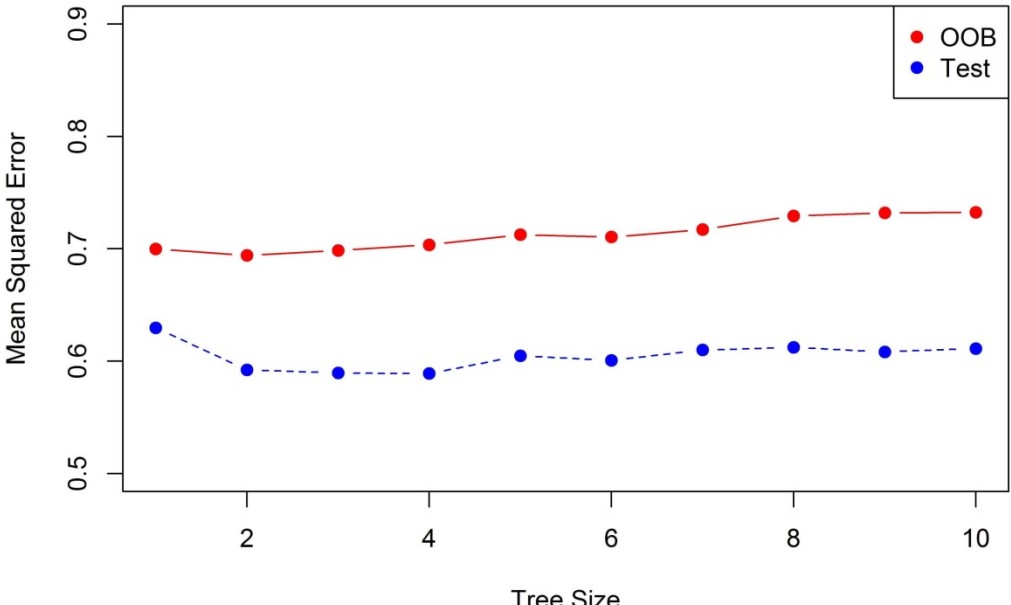

**Figure 10: The OOB and test showed as a function of the number of terminal nodes in the regression trees.**





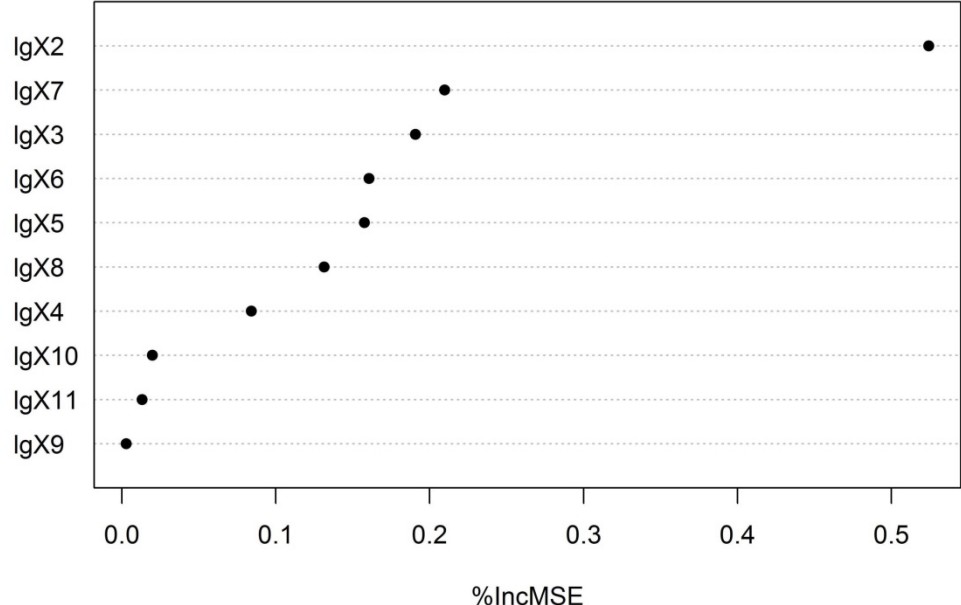

**Figure 11: Variable importance over all trees in the regression tree model for the ten predictors.**



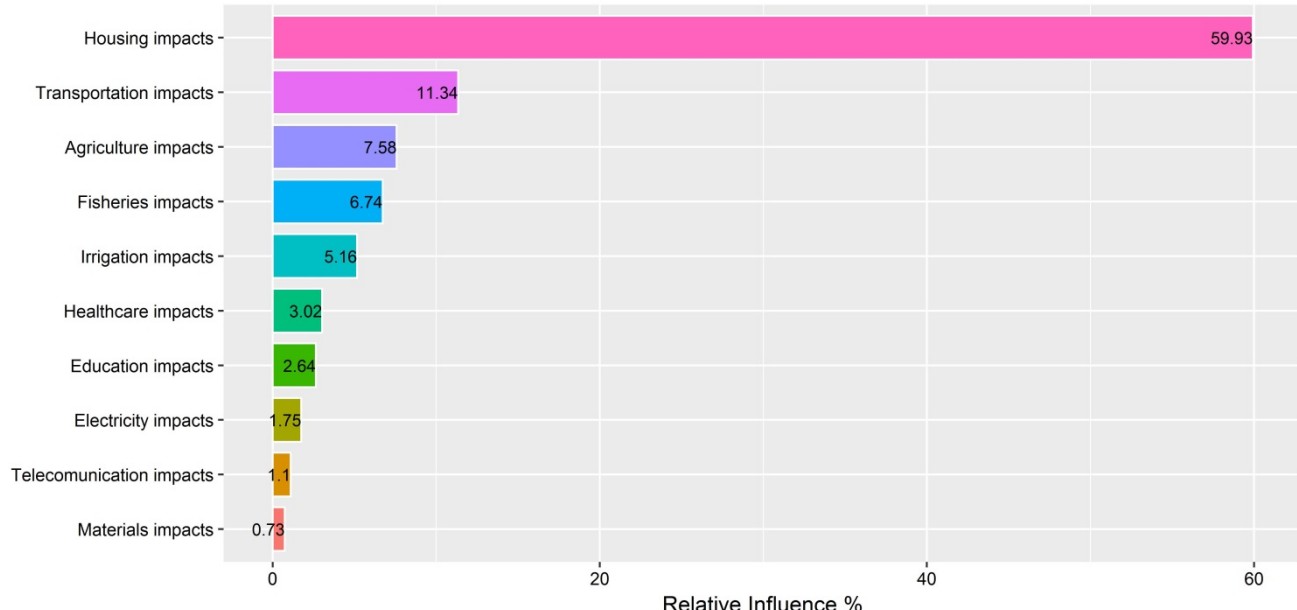

**Figure 12: Variable importance plot of flood damage impacts on fatalities from boosting technique.**





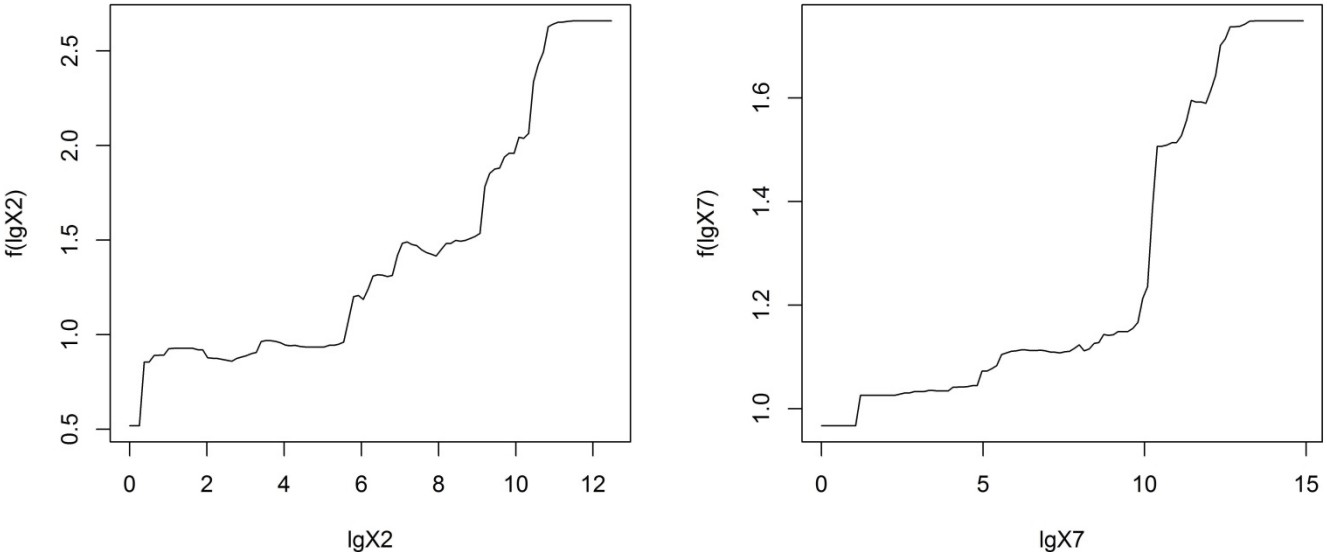

**Figure 13: Partial dependence plots for the most two important variables, housing impacts (lgX2) and transportation impacts (lgX7).**



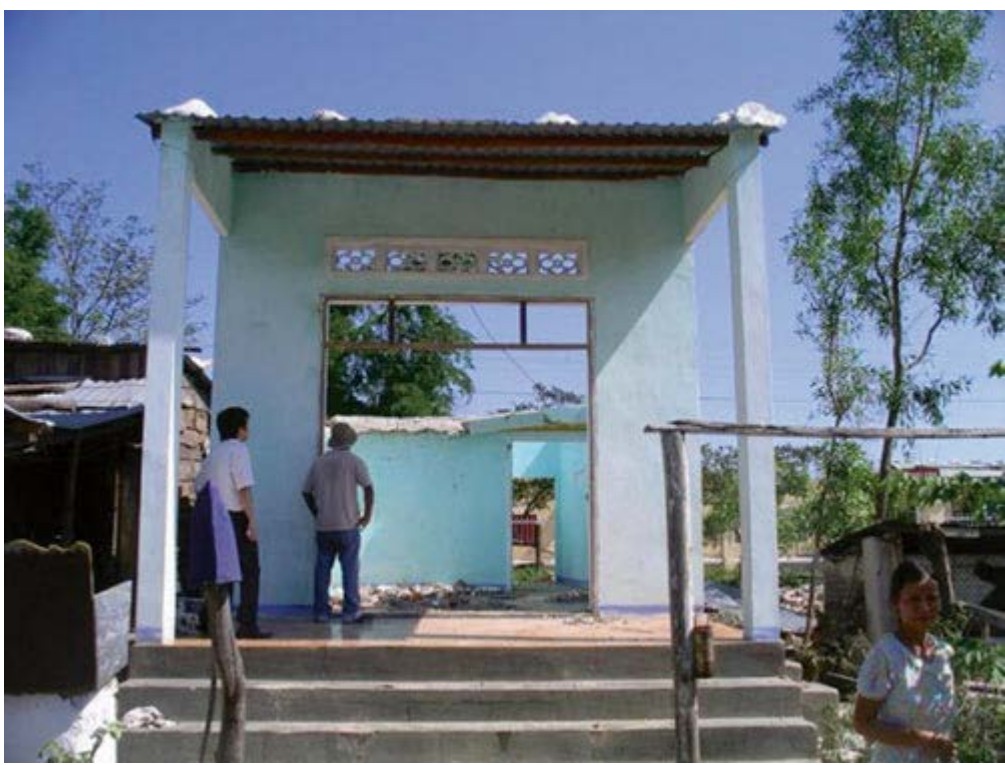

**Figure 14: A house collapsed by 2006 typhoon Xangsane in Central Vietnam (Tran, 2016).**





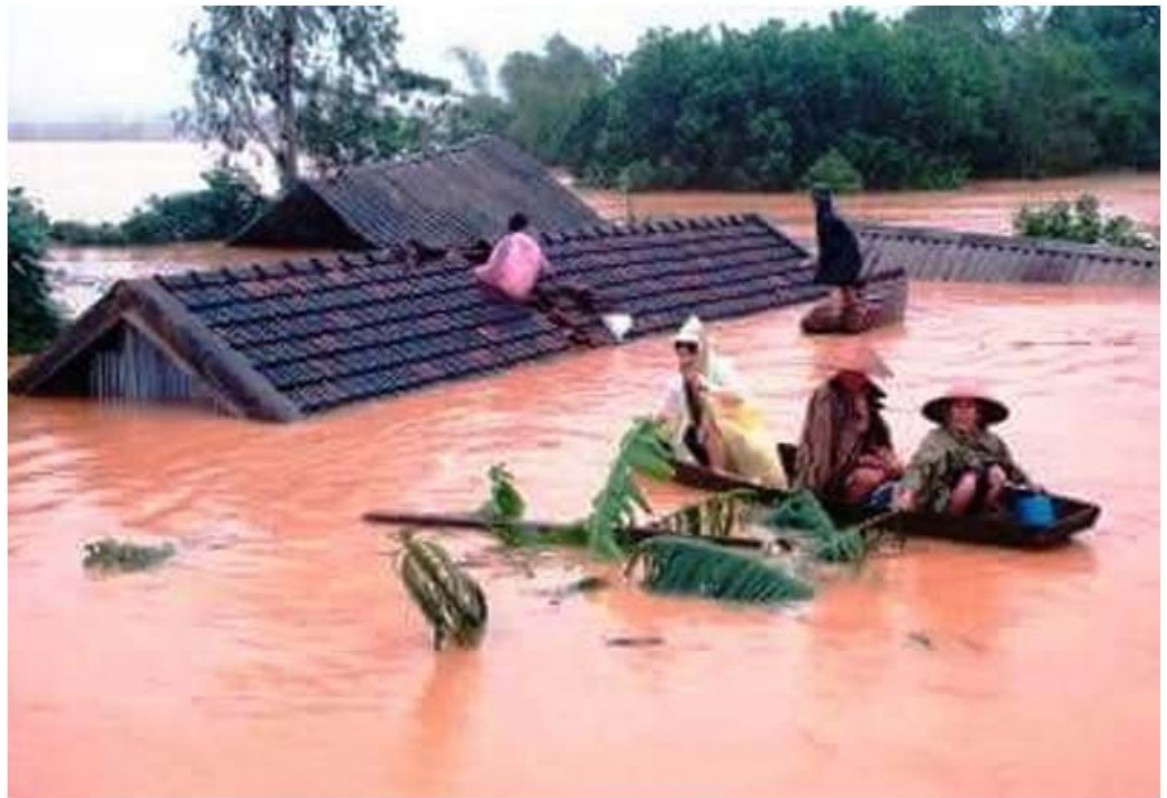

Figure 15: A person must lift roof tile to escape from flooding in Central Vietnam in November 2016 flood event (source:
http://phongchongthientai.vn/tin-tuc/mua-lu-gay-thiet-hai-tai-cac-tinh-mien-trung/-c3222.html).



**Table 1: 63 provinces and regions in Vietnam**

| Main regions | Regions | Provinces |
|---|---|---|
| North | Northwest (Tây Bắc Bộ) | Phu Tho, Ha Giang, Tuyen Quang, Cao Bang, Bac Kan, Thai Nguyen, Lang Son, Bac Giang, Quang Ninh |
| | Northeast (Đông Bắc Bộ) | Hoa Binh, Son La, Dien Bien, Lai Chau, Lao Cai, Yen bai |
| | Red River Delta (Đồng Bằng Sông Hồng) | Vinh Phuc, Ha Noi, Bac Ninh, Ha Nam, Hung Yen, Hai Duong, Hai Phong, Thai Binh, Nam Dinh, Ninh Binh |
| Central | North Central Coast (Bắc Trung Bộ) | Thanh Hoa, Nghe An, Ha Tinh, Quang Binh, Quang Tri, Thua Thien Hue |
| | South Central Coast (Nam Trung Bộ) | Da Nang, Quang Nam, Quang Ngai, Binh Dinh, Phu Yen, Khanh Hoa, Ninh Thuan, Binh Thuan |
| | Central Highlands (Tây Nguyên) | Kon Tum, Gia Lai, Dak Lak, Dak Nong, Lam Dong |
| South | Southeast (Đông Nam Bộ) | Thanh Pho Ho Chi Minh, Ba Ria Vung Tau, Binh Duong, Binh Phuoc, Dong Nai, Tay Ninh |
| | Mekong Delta (Đồng Bằng Sông Cửu Long) | An Giang, Bac Lieu, Ben Tre, Ca Mau, Can Tho, Dong Thap, Hau Giang, Kien Giang, Long An, Soc Trang, Tien Giang, Tra Vinh, Vinh Long |



**Table 2 Description of the 11 variables with 1701 observations for each variable**

| Impacts | Explanatory variables | Unit | Variables |
|---|---|---|---|
| Fatalities | Numbers of death and missing people | | X1a |
| Housing | Numbers of houses collapsed and washed away, and numbers of houses flooded and damaged | | X2 |
| Education | Numbers of classrooms collapsed and washed away, and numbers of classrooms damaged | | X3 |
| Healthcare | Numbers of clinics collapsed and washed away, and numbers of clinics submerged and damaged | | X4 |
| Agriculture | Areas of paddy inundated, areas of farm produce submerged, damaged, areas of seeding submerged, areas of industrial tree lost, areas of industrial tree damaged, areas of sugarcane damaged, areas of planted forest damaged, areas of orchard damaged | hectare | X5 |
| Irrigation | Volumes of earth eroded, washed away, and redeposited; and volumes of rock eroded, washed away, and redeposited (of dykes, canals, and reservoirs) | cubic meter | X6 |
| Transportation | Volumes of earth eroded, washed away, and redeposited; and volumes of rock eroded, washed away, and redeposited (of roads and highways) | cubic meter | X7 |
| Fisheries | Areas of fish and shrimp feeding area destroyed (hectare) | hectare | X8 |
| Telecommunication | Numbers of telephone poles collapsed | | X9 |
| Electricity | Numbers of high voltage electric towers broken, and numbers of electric distribution poles broken | | X10 |
| Materials | Volumes of cement damaged, volumes of salt lost, volumes of clinker wetted, volumes of coal drifted, volumes of rush damaged, and volumes of fertiliser damaged | ton | X11 |