# Peer review of "Analysing flood fatalities in Vietnam using national disaster database and tree-based methods"

_Natural Hazards and Earth System Sciences, 2017_

## Referee Comment (RC1) · Anonymous Referee #1 · 16 Jun 2017

**Journal: NHESS**
**Title: Analysing flood fatalities in Vietnam using national disaster database and tree-based methods**
**Author(s): Chinh Luu et al.**
**MS No.: nhess-2017-155**
**MS Type: Research article**
**Special Issue: Damage of natural hazards: assessment and mitigation**

The objectives are (1) providing a comprehensive overview of flood fatalities in Vietnam, and (2) examining the damage-influencing variables (flood impacts) on flood fatalities. Accordingly, tree-based methods and DANA database were used.

Despite the objectives of the study which is interesting, the paper suffers from major shortcomings which prevent its publication. In what follows, major criticisms are first discussed, and then specific comments are supplied.

**Major criticisms**

I.  With respect to the two objectives, neither a comprehensive overview nor examining of the damage-influencing variables (flood impacts) is analysed. To this aim, I expected the authors to analyse the significance of hazard and vulnerability influencing parameters on flood fatalities. Even, the importance of this matter has been stated by the authors in line 7-9 of page 1. However, the authors simply performed an exposure analysis which does not really need a tree-based model. In other words, as described in Table 2, the authors have only considered some damaged, exposed sectors with some quantities (not some ranges of variation), and damage-influencing variables are neglected entirely.

On the other hand, It is obvious that a majority number of the people are usually trapped in houses or caught in roads (in the time of escaping). Then, a tree based model represents these two influenced sectors since the quantity of them are relatively high.

All in all, exposure assessment would not be a damage influencing analysis when the authors have neglected the vulnerability (physical, social or systemic) of affected objects and the hazard intensity parameters.

II.  To my understanding there are some methodological inconsistencies in the paper:

1.  For implementing the above comment, an event-based analysis is needed. The authors have not considered a scientific approach for distributing the cumulated number of fatalities (between 1989 and 2015) in each year or each region. In Page 1 line 15 and page 5 line 13, the authors have simply divided 14,927 fatalities to 27 years (between 1989 and 2018) and reached to 553 numbers of casualties which is not scientifically sound. Also, this number is not compatible with Figure 5 information. For that, they needed to

calculate the Average Annual Damage (AAD) based on the probability (return period) of each event and the extent of losses of that.

2. It is not clear that how the authors have calculated the information of Figure 7 (annual average of losses), without assessing the AAD explained earlier. If it again comes from a simple division, it is not scientifically correct. Also, this figure suffers from several problems (e.g. incomplete caption; wrong axis label "SSC instead of SCC"; unnatural distribution of standard errors; incorrect length of bar charts for SE and NE which are equal to 2 and 5 respectively), and its information is not compatible with Fig.5.

3. Authors should describe their methodological steps chronologically to avoid confusion. There are many examples of the information which are represented in an inappropriate and unrelated section or repeated several times.

4. Discussion and Conclusion parts, as the most important sections of each study, should be rewritten entity. In the presented format, the discussion part is a repetition of previous materials, and the conclusion part does not represent any outcome, finding, or contribution.

5. The main application of out-of-bag (OOB) data, exploring the feature importance, is not used in this study. Then, what was the advantage of using this technique besides cross-validation approach?

6. It is obvious, and it has been shown before that Bagging and Random Forests represent more stable and accurate outcome. Consequently, I expected that the authors use MSE and cross-validation test for choosing a tree with the most appropriate number of nodes. However, it is not clear that why the authors have selected a tree with seven leaves (shown in Figure 9), while its error in Figure 10 is more than other sizes of trees.

7. In Page 10 line 26, the authors have mentioned that "The results showed that the tree-based models were validated and applicable". In this study validation of the model is not discussed and the study includes only some error estimations.

8. Results of Figure 2 are not compatible with the findings of the grown trees. In Figure 2, most of the sectors have a high (more than 0.5) correlation coefficient with the number of fatalities. This matter reflects the issue of the inaccurate inputs explained in section I.

9. Based on the explanations of Table 2, the relationship of some categories like "irrigation, telecommunication, electricity and material categories" with the number of fatalities is not clear. Are they related?!

III.     Transformation of the empirical data needs more clarification.

**Specific comments**

- Page 1, Line 10: "The number of fatalities is the most important indicator in flood risk assessment." Do you have any reference for justifying this statement?
- It is not clear enough that how we can use Figure 9 for predicting the number of fatalities. In this Figure, I do not understand that what "the small number on the top

of each rectangle" is for, and what does the number at the upper line of each rectangle mean? All of them need some explanations. Also, why the summation of the percentages of the end nodes (leaves) is 102%?!!!

- In Figure 9, there are considerable differences in the number of data related to each end node (625 in the first node and 44 in the fifth node). It shows that the tree is not grown soundly and the pruning technique is not hired perfectly.

- Page 6, Line 22: "The cross-validation procedure was undertaken to ensure that the parameter estimation and model generation of regression trees, bagging, random forests and boosting are **entirely independent of the test data**." Cross-validation test does not generate an independent dataset. The authors need to use data collected from another event if they are interested in testing the transferability and applicability of the model.

- Page 8 Line 5: "*Cross-validation method was operated to select the most accurate tree-based technique and to **check the data validation***". Cross-validation is a technique for validation of model compared to the real damage data. It is not an approach for checking the validation of the data.

- There is some information that does not have any contribution to the objectives of this study and the authors need to delete them such as Page 11 L 8-12; total injured people in Figure 5; Figure 14; Figure 15; and Table 1.

- Figures need some improvement (e.g. caption of Figure 2 needs more explanations about variables, presentation of Figure 3, SSC should be replaced by SCC in axis label of Figure 6, and NE should be replaced by NW in Figure 6 caption)

- Page 6 Line 7: "*Tree-based methods are supervised learning algorithms. The methodology of these methods is based on classification and regression tree (CART) of Breiman et al. (1984).*" Classification and regression tree (CART) is not the only algorithm of tree-based methods. Trees can be grown based on different algorithms such as RETIS (Karalic & Cestnik, 1991), M5 (Quinlan, 1992), REPTree, Random Tree, or CART (Breiman et. al., 1984).

---

## Referee Comment (RC2) · Anonymous Referee #2 · 23 Jun 2017

The manuscript deals with the analysis of flood fatalities in Vietnam using a national disaster database (DANA) and tree based methods. Despite the topic could be interesting, I would not recommend this paper for publication in NHESS. The fundamental problem relates to the objective of examining damage-influencing variables on flood fatalities using the DANA dataset, which only includes some damaged quantities for different sectors (e.g. number of collapsed houses, schools, hospitals, etc.), without any further information on hazard and vulnerability parameters. As a consequence, without such information, the results of the study are rather obvious (the number of fatalities is related to number of collapsed houses and roads), so it is hard to me to see the scientific contribution of the work.

Specific comments:

[Figure]

- In the abstract the authors say that "The findings allow us to make recommendations for government policies on improving housing quality for the poor in flood-prone areas in Vietnam", however a discussion on this point is lacking in the paper (except few lines in the conclusions).

- The authors state that "The study contributes a method to analyse the national disaster database, provides a substantial insight in flood-related fatalities in Vietnam and offers a valuable application for other Asian countries" (page 3, lines 12-13). The use of tree based methods is not new and many papers examining disaster damage data exist in the literature. In addition, considering the type of data analyzed in the paper, which are the "substantial insights" provided?

- Section 2 "Disaster damage data". This section should be improved providing more details regarding the DANA database.

- I would suggest to entirely rewrite sections 4, 5 and 6, avoiding repetition of concepts throughout the paper. In the present form, the paper lacks of a real discussion section, which is now just a repetition of the results presented in a previous section.

- The conclusion section is vague and it should be improved. The authors state that "This study can contribute to the body of flood hazard knowledge by analysing and reporting on flood fatalities in Vietnam" (page 11, lines 27-28). What is this contribution? In addition, your analysis did not include any information on flood hazard.

- The results of the grown trees seem to be not compatible with the results of Figure 2, where I can see other parameters having correlation coefficients comparable to the one observed for lgX7.

- Caption of Figure 9 should be improved.

Minor comments:

- Page 2, line 13: [. . .] "and to a lesser extent IN developing ones".

- Page 6, line 19: [. . .] "was performed to validATE".

- Page 11, line 23: "This paper proposed an approach [. . .]": the approach is not new, you should write instead that "This paper used an approach [. . .]".
* * *

---

## Author Comment (AC1) · 23 Jun 2017

**Response to comments of Reviewer 1:**

We sincerely thank the reviewer for taking time to read our work and give us many helpful and detailed comments and suggestions. Here we would like to address the reviewer's concerns as follows:

**Major criticisms**

1. *"With respect to the two objectives, neither a comprehensive overview nor examining of the damage-influencing variables (flood impacts) is analysed. To this aim, I expected the authors to analyse the significance of hazard and vulnerability influencing parameters on flood fatalities. Even, the importance of this matter has been stated by the authors in line 7-9 of page 1. However, the authors simply performed an exposure analysis which does not really need a tree-based model. In other words, as described in Table 2, the authors have only considered some damaged, exposed sectors with some quantities (not some ranges of variation), and damage-influencing variables are neglected entirely.*
   *On the other hand, It is obvious that a majority number of the people are usually trapped in houses or caught in roads (in the time of escaping). Then, a tree based model represents these two influenced sectors since the quantity of them are relatively high.*
   *All in all, exposure assessment would not be a damage influencing analysis when the authors have neglected the vulnerability (physical, social or systemic) of affected objects and the hazard intensity parameters."*

Response: The reviewer is correct to point out that this paper "performed an exposure analysis", and "considered some damages", and "exposure assessment would not be a damage influencing analysis". Upon reflection, we realise that we used an inappropriate term, "damage-influencing variables", in our writing. It should be "flood impact attributes" or "flood damage attributes" instead.

Some of the limitations that are highlighted relate to the available data in the national disaster database of Vietnam, DANA. Our paper was limited to the analysis of records of direct damage data, as the reviewer has pointed out.

The DANA data was previously available at http://www.ccfsc.gov.vn. However, the website changed to http://phongchongthientai.vn/default.asp since 2014. A part of the database can be found in this address http://www.desinventar.net/DesInventar/profiletab.jsp?countrycode=vnm&continue=y. We collected the data from the Central Steering Committee for Natural Disaster Prevention and Control of Vietnam with over 200 data cards. Due to the lack of studies using DANA and indeed the lack of studies on flood fatalities in Vietnam, we responded these gaps and assessed the relationship of flood impact attributes to flood fatalities in Vietnam.

The available database only includes direct flood damages, so we cannot analyse physical or social vulnerability or build a predictive model for flood fatalities using this data. We cited 4 papers using this method to analyse the significance of hazard and vulnerability influencing parameters (Merz et al., 2013; Chinh et al., 2015; Hasanzadeh Nafari et al., 2016; Wagenaar et al., 2017). We accessed another application of this method, which is to measure the importance of variables with random forests and boosting techniques. We aimed to assess the relative influence of flood impact attributes on flood fatalities.

The reviewer is correct to point out that "the authors simply performed an exposure analysis which does not really need a tree-based model". We aimed to assess the variable importance or relative influencing of flood damage attributes on flood fatalities instead of building a predictive model. The limitation is due to the data we have. We would like to modify the paper in analysing the variable importance of flood damage attributes on flood fatalities using linear regression, regression tree,

random forests, conditional variable importance for random forests of Strobl et al. (2008) and boosting techniques. The discussion section will focus on comparing the results of these techniques.

2. *"To my understanding there are some methodological inconsistencies in the paper:*
*1. For implementing the above comment, an event-based analysis is needed. The authors have not considered a scientific approach for distributing the cumulated number of fatalities (between 1989 and 2015) in each year or each region. In Page 1 line 15 and page 5 line 13, the authors have simply divided 14,927 fatalities to 27 years (between 1989 and 2018) and reached to 553 numbers of casualties which is not scientifically sound. Also, this number is not compatible with Figure 5 information. For that, they needed to calculate the Average Annual Damage (AAD) based on the probability (return period) of each event and the extent of losses of that."*

Response: While the reviewer makes a valid criticism, we do not have detailed data on the probability of all events between 1989 and 2015 in DANA. The recorded data is only available for direct damages. We therefore intend to remove the average numbers in the paper, or report in a more qualified manner.

*"2. It is not clear that how the authors have calculated the information of Figure 7 (annual average of losses), without assessing the AAD explained earlier. If it again comes from a simple division, it is not scientifically correct. Also, this figure suffers from several problems (e.g. incomplete caption; wrong axis label "SSC instead of SCC"; unnatural distribution of standard errors; incorrect length of bar charts for SE and NE which are equal to 2 and 5 respectively), and its information is not compatible with Fig.5."*

Response: We agree with the reviewer's comments but we do not have data on the probability of all events between 1989 and 2015 in DANA, so intend to remove Figure 7. The average numbers of Figure 7 resulted from the division of the total number of a region (in Figure 5) to the number of provinces in a region, but the numbers of Figure 5 and Figure 7 are from one input data. The typo error will be fixed according to the reviewer's comment. Our error on "incorrect length of bar charts for SE and NE" is due to the rounded numbers we made.

*"3. Authors should describe their methodological steps chronologically to avoid confusion. There are many examples of the information which are represented in an inappropriate and unrelated section or repeated several times."*

Response: We agree that it is important to "describe their methodological steps chronologically to avoid confusion". We will revise this section thoroughly.

*"4. Discussion and Conclusion parts, as the most important sections of each study, should be rewritten entity. In the presented format, the discussion part is a repetition of previous materials, and the conclusion part does not represent any outcome, finding, or contribution."*

Response: It is true that Discussion and Conclusion parts are the most important sections of each study. We acknowledge that these parts might be rewritten.

*"5. The main application of out-of-bag (OOB) data, exploring the feature importance, is not used in this study. Then, what was the advantage of using this technique besides cross-validation approach?"*

Response: The Mean Square Residual (MSR) and percentage variance explained are based on out-of-bag (OOB) estimates to get honest error estimates. In our case, the results of OOB error in Figure 10, which were estimated with a function of $m_{try}$ (the number of selected variables) from 1 to 10, can be used to choose the most suitable $m_{try}$. The $m_{try}$ = 3 seems to be the best for OOB error in Figure 10.

However, the result in Figure 10 showed that the difference of MSEs is quite small, $m_{try}$ can be selected from 1 to 10.

*"6. It is obvious, and it has been shown before that Bagging and Random Forests represent more stable and accurate outcome. Consequently, I expected that the authors use MSE and cross-validation test for choosing a tree with the most appropriate number of nodes. However, it is not clear that why the authors have selected a tree with seven leaves (shown in Figure 9), while its error in Figure 10 is more than other sizes of trees."*

Response: We only aimed to assess the variable importance or relative influencing of flood damage attributes on flood fatalities using random forests and boosting techniques, so we did not present about selecting the number of nodes. We will add the cross-validation for pruning the tree in the paper. The result with cross-validation is shown in the following figure. The pruning tree with 7 terminal nodes (leaves) is the best.

[Figure]

The results of OOB and test errors in Figure 10, which were estimated with a function of $m_{try}$ (the number of selected variables) from 1 to 10, were used to choose the most suitable $m_{try}$.

*"7. In Page 10 line 26, the authors have mentioned that "The results showed that the tree-based models were validated and applicable". In this study validation of the model is not discussed and the study includes only some error estimations.*

Response: We acknowledged the shortcomings in our study based on the reviewer's comments. We will revise the whole paper. We would like to explain steps we used as follows:

1. We collected a data set
2. The original samples were randomly divided into two groups, train and test datasets with equal size.
3. A model was developed in the train dataset.
4. The model was validated using the testing dataset.
5. Mean Squared Error (MSE) was used to evaluate the performance of model.

We will apply linear regression model, regression tree, bagging, random forests and boosting and compare the results among models in the revised paper.

*"8. Results of Figure 2 are not compatible with the findings of the grown trees. In Figure 2, most of the sectors have a high (more than 0.5) correlation coefficient with the number of fatalities. This matter reflects the issue of the inaccurate inputs explained in section I."*

Response: We used the DANA data for the input data. It consists of direct flood impact attributes as described in Table 2. The observed data are 27 years, and we compiled the data by 63 provinces in Vietnam, so we have 1701 observations. The random forests and boosting techniques are used to assess the relative influencing of flood impact attributes on flood fatalities. Bi (2012) mentioned about advantages of Breiman's random forests to determine the variable importance of correlated predictors; and Strobl et al. (2008) said that conditional variable importance for random forests is suitable for high correlated predictors. We will add the analysis and results of conditional variable importance in the paper.

*"9. Based on the explanations of Table 2, the relationship of some categories like "irrigation, telecommunication, electricity and material categories" with the number of fatalities is not clear. Are they related?!"*

Response: It is true that the flood impact categories of electricity, telecommunication and material have quite small relative influence on flood fatalities with 1.75%, 1.1% and 0.73% respectively as the result in Figure 12. The result showed that they are not related to flood fatalities.

3. *"Transformation of the empirical data needs more clarification"*

Response: We agree that it is a very important point. Again, we had to depend on the quality of available data. The data used for analysis is flood damage data, so it is random and contains many zero values. Some observations contain all zero values if no storms or floods occurred. Besides, the transformed data performed much better than the original ones. We would like to provide R code for the original data and the transformed data with bagging technique as follows:

```
> set.seed(1)
> train=sample(1:nrow(dat2),nrow(dat2)/2)
> test=-train
> bag.fatal1=randomForest(X1a~X2+X3+X4+X5+X6+X7+X8+X9+X10+X11, data=dat2, subset=train, mtry=10, importance=TRUE)
> bag.fatal1

Call:
 randomForest(formula = X1a ~ X2 + X3 + X4 + X5 + X6 + X7 + X8 +   X9 + X10 + X11, data = dat2,
mtry = 10, importance = TRUE,      subset = train)
               Type of random forest: regression
                     Number of trees: 500
No. of variables tried at each split: 10

         Mean of squared residuals: 490.1336
                   % Var explained: 15.99
> bag.fatal=randomForest(lgX1a~lgX2+lgX3+lgX4+lgX5+lgX6+lgX7+lgX8+lgX9+lgX10+lgX11, data=dat2, subset=train, mtry=10, importance=TRUE)
> bag.fatal

Call:
 randomForest(formula = lgX1a ~ lgX2 + lgX3 + lgX4 + lgX5 + lgX6 +   lgX7 + lgX8 + lgX9 + lgX10
+ lgX11, data = dat2, mtry = 10,      importance = TRUE, subset = train)
               Type of random forest: regression
                     Number of trees: 500
No. of variables tried at each split: 10

         Mean of squared residuals: 0.616645
                   % Var explained: 62.8
```

The results showed that the transformed data performed much better than the original one when comparing their mean squared residuals and percentage variance explained.

**Specific comments**

1. *"Page 1, Line 10: "The number of fatalities is the most important indicator in flood risk assessment." Do you have any reference for justifying this statement?"*

   Response: We would like to fix this by "The number of fatalities is an important indicator in flood risk assessment".

2. *"It is not clear enough that how we can use Figure 9 for predicting the number of fatalities. In this Figure, I do not understand that what "the small number on the top of each rectangle" is for, and what does the number at the upper line of each rectangle mean? All of them need some explanations. Also, why the summation of the percentages of the end nodes (leaves) is 102%?!!!"*

   Response: It is true that we cannot use Figure 9 for predicting the number of fatalities. With the data we have, we can only assess the variable importance of predictors on flood fatalities.

   We would like to explain the meaning of "the small number on the top of each rectangle" and "the number at the upper line of each rectangle" by the following R code. They are the order of nodes and the yval or fitted value at terminal nodes with the green and red values respectively.

```
> print(tree.fatal1)
n= 1701
node), split, n, deviance, yval
      * denotes terminal node

 1) root 1701 2901.76100 1.1099730
   2) lgX2< 6.914713 1228 1045.49500 0.6011133
     4) lgX2< 0.3465736 625   223.62820 0.1661002 *
     5) lgX2>=0.3465736 603   581.00640 1.0519970
      10) lgX7< 10.06186 520   417.41590 0.9207179 *
      11) lgX7>=10.06186 83    98.48223 1.8744720 *
   3) lgX2>=6.914713 473   712.76150 2.4310720
     6) lgX2< 10.40862 351   422.89200 2.0650810
      12) lgX7< 10.53584 206   270.50130 1.6839450
        24) lgX2< 9.405371 162   162.41080 1.4828570 *
        25) lgX2>=9.405371 44    77.42142 2.4243140 *
      13) lgX7>=10.53584 145    79.95262 2.6065580 *
     7) lgX2>=10.40862 122   107.58520 3.4840460 *
```

   We had an error regarding "the summation of the percentages of the end nodes (leaves) is 102%?!!!" because the used package rounded the percentage numbers. We have to use decimal numbers to draw the tree. We would like to demonstrate in another tree as follows:

[Figure]

3. *"In Figure 9, there are considerable differences in the number of data related to each end node (625 in the first node and 44 in the fifth node). It shows that the tree is not grown soundly and the pruning technique is not hired perfectly.*

    Response: We agree with the reviewer's comment on the grown tree because it depends on the quality of the data we have. We ran cross-validation for pruning the tree, and the result showed that the pruning tree with 7 terminal nodes is the best (the figure is shown in response to the reviewer's comment No. 6 above).

4. *"Page 6, Line 22: "The cross-validation procedure was undertaken to ensure that the parameter estimation and model generation of regression trees, bagging, random forests and boosting are entirely independent of the test data." Cross-validation test does not generate an independent dataset. The authors need to use data collected from another event if they are interested in testing the transferability and applicability of the model".*

    Response: We thank the reviewer for pointing out this incorrect explanation. We will rewrite these sentences. We would like to explain the steps we used:

    1. We collected a data set.

    2. The original samples were randomly divided into two groups, train and test datasets with equal size.

    3. A model was developed in the train dataset.

    4. The model was validated using the testing dataset.

    5. Mean Squared Error (MSE) was used to evaluate the performance of the model.

    We used train subset to run the bagging, random forests and boosting techniques. We used cross-validation to validate the models. The test dataset was used to validate. We would like to show R code we used with bagging technique as follows:

```
> train=sample(1:nrow(dat2),nrow(dat2)/2)
> test=-train
> bag.fatal=randomForest(lgX1a~lgX2+lgX3+lgX4+lgX5+lgX6+lgX7+lgX8+lgX9+lgX10+lgX11,data=dat2,subs
et=train,mtry=10,importance=TRUE)
> bag.fatal

Call:
 randomForest(formula = lgX1a ~ lgX2 + lgX3 + lgX4 + lgX5 + lgX6 +  lgX7 + lgX8 + lgX9 + lgX10 +
lgX11, data = dat2, mtry = 10,      importance = TRUE, subset = train)
               Type of random forest: regression
                     Number of trees: 500
No. of variables tried at each split: 10

        Mean of squared residuals: 0.616645
                  % Var explained: 62.8
> yhat.bag=predict(bag.fatal,newdata = dat2[-train,])
> fatal.test=dat2[-train,"lgX1a"]
> mean((yhat.bag-fatal.test)^2)
[1] 0.7284176
```

    The test dataset MSE associated with the bagged regression tree is 0.73.

    We will compare MSEs among models (linear regression, regression tree, bagging, random forests and boosting) and add discussions about results of MSEs in the revised paper.

5. *"Page 8 Line 5: "Cross-validation method was operated to select the most accurate tree-based technique and to check the data validation". Cross-validation is a technique for validation of model compared to the real damage data. It is not an approach for checking the validation of the data."*

Response: We acknowledged our incorrect explanations. In our case, we used cross-validation to validate the model, following the five steps in the response above.

6. *"There is some information that does not have any contribution to the objectives of this study and the authors need to delete them such as Page 11 L 8-12; total injured people in Figure 5; Figure 14; Figure 15; and Table 1."*

Response: We agree with the reviewer's comment. We will remove these items in the paper.

7. *"Figures need some improvement (e.g. caption of Figure 2 needs more explanations about variables, presentation of Figure 3, SSC should be replaced by SCC in axis label of Figure 6, and NE should be replaced by NW in Figure 6 caption)"*

Response: We thank the reviewer for detailed comments. We will improve these figures in the paper.

8. *Page 6 Line 7: "Tree-based methods are supervised learning algorithms. The methodology of these methods is based on classification and regression tree (CART) of Breiman et al. (1984)." Classification and regression tree (CART) is not the only algorithm of tree-based methods. Trees can be grown based on different algorithms such as RETIS (Karalic & Cestnik, 1991), M5 (Quinlan, 1992), REPTree, Random Tree, or CART (Breiman et. al., 1984).*

Response: The reviewer's comment broadened our thinking. We will read more about the other algorithms.

---

## Author Comment (AC2) · 2 Jul 2017

**Response to comments of Reviewer 2:**

We sincerely thank the reviewer for taking time to read our work and give us helpful and detailed comments. Here we would like to address the reviewer's concerns as follows:

1. *"The manuscript deals with the analysis of flood fatalities in Vietnam using a national disaster database (DANA) and tree based methods. Despite the topic could be interesting, I would not recommend this paper for publication in NHESS. The fundamental problem relates to the objective of examining damage-influencing variables on flood fatalities using the DANA dataset, which only includes some damaged quantities for different sectors (e.g. number of collapsed houses, schools, hospitals, etc.), without any further information on hazard and vulnerability parameters. As a consequence, without such information, the results of the study are rather obvious (the number of fatalities is related to number of collapsed houses and roads), so it is hard to me to see the scientific contribution of the work."*

   Response: We support the reviewer's assertion that "The fundamental problem relates to the objective of examining damage-influencing variables on flood fatalities using the DANA dataset, which only includes some damaged quantities for different sectors (e.g. number of collapsed houses, schools, hospitals, etc.), without any further information on hazard and vulnerability parameters". Some of the limitations that are highlighted relate to the available data in the national disaster database of Vietnam, DANA. Our paper was limited to the analysis of records of direct damage data, as the reviewer has pointed out.

   The DANA data was previously available at http://www.ccfsc.gov.vn. However, the website changed to http://phongchongthientai.vn/default.asp since 2014. A part of the database can be found in this address http://www.desinventar.net/DesInventar/profiletab.jsp?countrycode=vnm&continue=y. We collected the data from the Central Steering Committee for Natural Disaster Prevention and Control of Vietnam with over 200 data cards from 1989 to 2015. Due to the lack of studies using DANA and indeed the lack of studies on flood fatalities in Vietnam, we responded these gaps and assessed the relationship of flood impact attributes to flood fatalities in Vietnam.

   We agree with the reviewer that "As a consequence, without such information, the results of the study are rather obvious (the number of fatalities is related to number of collapsed houses and roads), so it is hard to me to see the scientific contribution of the work". However, we still think that the study can have some small contributions with the objective of analysing the national disaster database of Vietnam (DANA). The results can be a validation for the variable important analysis using tree-based methods in this study. In addition, the analysis using boosting technique can provide a measurement of variable importance (of flood impact attributes on flood fatalities) and partial dependence plots for the variables as in Figure 12 and Figure 13.

**Specific comments**

2. *"In the abstract the authors say that "The findings allow us to make recommendations for government policies on improving housing quality for the poor in flood-prone areas in Vietnam", however a discussion on this point is lacking in the paper (except few lines in the conclusions)"*

   Response: We agree that it is an important aspect. We had some discussion on the housing factor and housing quality in Discussion section in the last paragraph on page 10 and the first paragraph on page 11 with the illustration of Figure 14 and Figure 15.

3. *"The authors state that "The study contributes a method to analyse the national disaster database, provides a substantial insight in flood-related fatalities in Vietnam and offers a valuable application for other Asian countries" (page 3, lines 12-13). The use of tree based methods is not new and many*

*papers examining disaster damage data exist in the literature. In addition, considering the type of data analyzed in the paper, which are the "substantial insights" provided?"*

Response: We agree with the reviewer that "The use of tree based methods is not new and many papers examining disaster damage data exist in the literature". We would like to mean that it is the first time this method is used to analyse a national disaster database. Besides Vietnam, many other countries also have their national disaster database, such as 94 national disaster databases are stored and available at http://www.desinventar.net/DesInventar/index.jsp. To our best knowledge, there have been no long-term empirical studies on the application of machine learning algorithms in investigating the relative importance of variables on flood fatalities.

4. *"Section 2 "Disaster damage data". This section should be improved providing more details regarding the DANA database."*

Response: We agree that it is an important point. We will revise this section thoroughly.

5. *"I would suggest to entirely rewrite sections 4, 5 and 6, avoiding repetition of concepts throughout the paper. In the present form, the paper lacks of a real discussion section, which is now just a repetition of the results presented in a previous section."*

Response: We thank the reviewer for this suggestion. We will revise these sections according to the reviewer's comment.

6. *"The conclusion section is vague and it should be improved. The authors state that "This study can contribute to the body of flood hazard knowledge by analysing and reporting on flood fatalities in Vietnam" (page 11, lines 27-28). What is this contribution? In addition, your analysis did not include any information on flood hazard."*

Response: We acknowledge that the conclusion section needs improving. We would like to mean that this study can have some small contributions in analysing the national disaster database of Vietnam (DANA). Due to the lack of studies using DANA and indeed the lack of studies on flood fatalities in Vietnam, we responded these gaps and assessed the relationship of flood impact attributes to flood fatalities in Vietnam.

The reviewer is correct to point out that our analysis did not include any information on flood hazard. We only analysed flood damage attributes that are available in the national disaster database.

7. *"The results of the grown trees seem to be not compatible with the results of Figure 2, where I can see other parameters having correlation coefficients comparable to the one observed for lgX7."*

Response: It is true that Figure 2 showed that the predictors are highly correlated. Bi (2012) mentioned about advantages of Breiman's random forests to determine the variable importance of correlated predictors; and Strobl et al. (2008) said that conditional variable importance for random forests is suitable for high correlated predictors. We will add the analysis using conditional variable importance for random forest in the paper.

8. *"Caption of Figure 9 should be improved"*

Response: We thank the reviewer's comment. We would like to demonstrate in another tree as follows:

[Figure]

**Minor comments:**

9. *"Page 2, line 13: [. . .] "and to a lesser extent IN developing ones""*

    Response: We thank reviewer's comment. We will fix this error.

10. *"Page 6, line 19: [. . .] "was performed to validATE""*

    Response: We thank reviewer's comment. We will fix this error.

11. *"Page 11, line 23: "This paper proposed an approach [. . .]": the approach is not new, you should write instead that "This paper used an approach [. . .]"*

    Response: We will rewrite as the reviewer's comment.